# The Clinical Significance of LDL-Cholesterol on the Outcomes of Hemodialysis Patients with Acute Coronary Syndrome

**DOI:** 10.3390/medicina59071312

**Published:** 2023-07-15

**Authors:** Keren Cohen-Hagai, Sydney Benchetrit, Ori Wand, Ayelet Grupper, Moshe Shashar, Olga Solo, David Pereg, Tali Zitman-Gal, Feras Haskiah, Daniel Erez

**Affiliations:** 1Department of Nephrology and Hypertension, Meir Medical Center, Kfar Saba 44281, Israel; 2Faculty of Medicine, Tel Aviv University, Tel Aviv 69978, Israeldaniel.erez@clalit.org.il (D.E.); 3Division of Pulmonary Medicine, Barzilai University Medical Center, Ashkelon 7830604, Israel; 4Faculty of Health Sciences, Ben-Gurion University of the Negev, Beer-Sheva 84105, Israel; 5Department of Nephrology and Hypertension, Tel Aviv Sourasky Medical Center, Tel Aviv 6423906, Israel; 6Department of Nephrology and Hypertension, Laniado Hospital, Netanya 4244916, Israel; 7Department of Anesthesiology, Pain and Intensive Care, Meir Medical Center, Kfar Saba 4428164, Israel; 8Department of Cardiology, Meir Medical Center, Kfar Saba 4428164, Israel; 9Department of Internal Medicine D, Meir Medical Center, Kfar Saba 4428164, Israel

**Keywords:** cardiovascular disease, dyslipidemia, LDL cholesterol, hemodialysis, chronic kidney disease

## Abstract

*Background and objectives*: Dyslipidemia is one of the most important modifiable risk factors in the pathogenesis of cardiovascular disease in the general population, but its importance in the hemodialysis (HD) population is uncertain. *Materials and Methods*: This retrospective cohort study includes HD patients hospitalized due to acute coronary syndrome (ACS) in the period 2015–2020 with lipid profile data during ACS. A control group with preserved kidney function was matched. Risk factors for 30-day and 1-year mortality were assessed. *Results*: Among 349 patients included in the analysis, 246 were HD-dependent (“HD group”). HD group patients had higher prevalence of diabetes, hypertension, and heart disease than the control group. At ACS hospitalization, lipid profile and chronic statin treatment were comparable between groups. Odds ratios for 30-day mortality in HD vs. control group was 5.2 (95% CI 1.8–15; *p* = 0.002) and for 1-year, 3.4 (95% CI 1.9–6.1; *p* <0.001). LDL and LDL < 70 did not change 30-day and 1-year mortality rates in the HD group (*p* = 0.995, 0.823, respectively). However, survival after ACS in HD patients correlated positively with nutritional parameters such as serum albumin (r = 0.368, *p* < 0.001) and total cholesterol (r = 0.185, *p* < 0.001), and inversely with the inflammatory markers C-reactive protein (CRP; r = −0.348, *p* < 0.001) and neutrophils-to-lymphocytes ratio (NLR; r = −0.181, *p* = 0.019). Multivariate analysis demonstrated that heart failure was the only significant predictor of 1-year mortality (OR 2.8, *p* = 0.002). LDL < 70 mg/dL at ACS hospitalization did not predict 1-year mortality in the HD group. *Conclusions*: Despite comparable lipid profiles and statin treatment before and after ACS hospitalization, mortality rates were significantly higher among HD group. While malnutrition–inflammation markers were associated with survival of dialysis patients after ACS, LDL cholesterol was not. Thus, our study results emphasize that better nutritional status and less inflammation are associated with improved survival among HD patients.

## 1. Introduction

Chronic kidney disease (CKD) is associated with a marked increase in incidence of mortality from cardiovascular disease (CVD) [1,2]. The annual mortality rate from CVD among patients with end-stage-kidney disease (ESKD) is high and estimated to be over 50% of the overall mortality rate [3,4,5]. 

A high level of low-density lipoprotein cholesterol (LDL-C) is a major risk factor for cardiovascular disease among the general population [6], and abnormalities in serum lipid levels associated with renal disease rank high among the factors responsible for accelerated atherosclerosis [7]. Approximately 20–40% of hemodialysis (HD) patients have elevated triglycerides and reduced high-density lipoprotein (HDL) cholesterol levels [8,9]. In addition, increased oxidized LDL-C and increased lipoprotein levels have been reported in up to 75% of HD patients [10]. A meta-analysis of randomized trials undertaken mainly in patients without CKD showed that statin therapy reduces the risk of major coronary events by about one-fifth for each 1 mmol/L reduction in LDL-C [11,12]. However, previous observational data on the effect of LDL-C levels in patients with ESKD yielded conflicting results [13,14,15]. It has been suggested that as estimated glomerular filtration rate (eGFR) falls below 30 mL/min, a cardiovascular pathophysiology other than accelerated atherosclerosis emerges; and additional variables contribute to the increased rate of cardiovascular events, including oxidative stress, endothelial dysfunction, vascular calcifications and stiffness, sympathetic overactivity [16,17,18], and malnutrition–inflammation atherosclerosis [19]. The role of inflammation as a risk factor for malnutrition in HD has been recognized in previous trials, and C-reactive protein (CRP) levels have been correlated with the inhibition of albumin synthesis in this particular population [20]. Several observational trials have noted that lower total cholesterol levels are associated with greater mortality in HD patients (“reverse epidemiology”) [13,21]. It has been suggested that this paradox results from reverse causation in which advanced cardiovascular disease leads to inflammation, wasting, and lower cholesterol levels, or a confounding effect that results in lower cholesterol levels and increased mortality [22]. In addition, the neutrophils-to-lymphocytes ratio (NLR), a simple biomarker of inflammation, has been shown to predict cardiovascular and all-cause mortality in HD patients [23].

Previous randomized controlled trials which assessed the impact of cholesterol-lowering treatment on cardiovascular outcomes in HD patients have also resulted in conflicting findings [24,25,26]. While the 4D study [24] and the AURORA study [26] did not detect significant benefits, the SHARP study [25] demonstrated a reduction in major cardiovascular endpoints (MACE) with cholesterol-lowering therapy in a group composed of dialysis and non-dialysis patients. However, the authors mentioned that the study did not have sufficient power to assess the effects on MACE separately in dialysis patients. 

Of note, the trials mentioned above did not specifically assess outcomes related to baseline cholesterol levels during acute coronary syndrome (ACS). It has been shown that high-dose statin treatment in ACS patients positively influences the balance between atherogenic and anti-atherogenic lipoproteins in the general population, leading to structural changes in the composition and stability of the atherosclerotic plaque, and producing anti-inflammatory effects and thus improving clinical outcomes [27].

Data regarding baseline cholesterol levels in HD patients during ACS and their relation to adverse cardiovascular outcomes are scarce. The objective of this study was to assess LDL-C levels in a group of HD patients presenting with ACS, and its association with all-cause mortality, compared to patients with normal renal function.

## 2. Materials and Methods

### 2.1. Study Design 

This retrospective observational study, conducted at Meir Medical Center, evaluated the clinical outcomes of patients hospitalized with a diagnosis of ACS during 2015–2020, with follow-up until 1 January 2021. 

Results are reported according to the STROBE statement guidelines.

ACS included all patients with a discharge diagnosis of unstable angina, ST-segment elevation myocardial infarction, or non-ST-segment elevation myocardial infarction. 

### 2.2. Study Population 

Our analysis consisted of chronic HD patients treated in the Dialysis Unit at Meir Medical Center from 2015 to 2020, who were hospitalized with a diagnosis of ACS. Chronic HD was defined as at least 3 months of dialysis treatment. 

Admission and discharge diagnoses were recorded by the attending physicians based on clinical, electrocardiographic, and biochemical criteria. Mortality data at 30 days and at 1 year post-hospitalization were determined from electronic medical records (EMR) and by the Israeli National Mortality Registry.

The control group was matched to the dialysis group based on age, sex, and hospitalization period. The control group was composed of patients hospitalized with ACS during the same period, who had eGFR > 60 mL/min. EMR were reviewed. Kidney function was assessed using eGFR based on the CKD epidemiology collaboration equation [28].

Patients < 18 years of age or missing EMR data or dialysis vintage < 3 months were excluded, as were subjects treated with PCSK9 inhibitors.

### 2.3. Variables 

Baseline demographic and clinical characteristics of the study patients, including age, sex, medical history of hypertension, diabetes, congestive heart failure, ischemic heart disease, atrial fibrillation, cerebrovascular accident, and peripheral vascular disease were retrieved from the EMR. Laboratory parameters at presentation, including white blood cells (WBC), hemoglobin, platelets, CRP, serum creatinine, urea, albumin, total cholesterol, LDL-C, HDL-C, and triglycerides, were also collected.

NLR was calculated as the ratio of neutrophils-to-lymphocytes from the differential white blood cell count routinely evaluated at the beginning of dialysis sessions. 

### 2.4. Ethical Considerations

The study was approved by the Institutional Ethics Committee in keeping with the principles of the Declaration of Helsinki. In accordance with Ministry of Health regulations, the Institutional Ethics Committee did not require written informed consent because data were collected anonymously from the EMR without active patient participation.

### 2.5. Statistical Analysis

Data are presented as numbers and percentages for nominal parameters and as means and standard deviations for continuous parameters. Differences between the study groups were analyzed using the chi-square test. Continuous variables were analyzed using the one-way ANOVA test. Spearman correlation was calculated to assess correlation between continuous variables. A multivariate logistic regression model including all significant variables in the univariate analysis was applied to estimate odds ratios. Multivariate models for predicting mortality were conducted using logistic regression analysis comparing 30-day and 1-year mortality rates by kidney function group (the control group with eGFR >60 mL/min served as the reference group). The HD group was compared to the control group. Kaplan–Meier survival curve analysis was performed in order to detect differences in survival of HD patients following ACS when comparing by LDL-C. *p* < 0.05 was considered statistically significant. 

Data were analyzed with SPSS Version 27 (IBM Corporation, Armonk, NY, USA).

## 3. Results

This retrospective analysis included 349 patients, among which 246 were defined as chronic dialysis patients prior to hospitalization for ACS (HD group). 

Baseline demographic and clinical characteristics of the patients are summarized in Table 1. HD patients had higher prevalence of diabetes, hypertension, heart failure, ischemic heart disease, peripheral vascular disease, and atrial fibrillation. They also had lower hemoglobin levels compared to the control group.

### 3.1. LDL-C in the HD Group 

LDL-C levels were normally distributed. Average and median LDL-C levels at admission were similar for the HD and control groups. (Table 1 and Figure 1). 

Eighty HD patients had LDL-C < 70 mg/dL (32.5%) versus 16 subjects (15.5%) in the control group (*p* < 0.01, Table 2). Despite a higher prevalence of LDL-C < 70 mg/dl in the HD group, rates of statin treatment before and after ACS hospitalization were comparable between groups (Table 2).

### 3.2. 30-Day Mortality

Among the 349 patients in this cohort, 47 died within 30-days after ACS admission. The 30-day mortality rate was higher in the HD group (43 patients, 17.5%) vs. the control group (four patients, 3.9%, *p* < 0.01). OR for 30-day mortality in the HD group was 5.2 (95% CI 1.8–15) vs. control group (Table 3). 

### 3.3. One-Year Mortality 

In the HD group, 99 patients (40.2%) died within 1 year as compared to 16.5% in the control group (*p* < 0.001). OR for 1-year mortality in the HD group was 3.4 (95% CI 1.9–6.1) vs. the control group (Table 3).

One-year mortality rates were similar for HD patients with LDL-C < 70 mg/dL: 33/80, 41.3%, and those with LDL-C ≥ 70 mg/dL, 66/166, 39.8%, *p* = 0.82. Similarly, low LDL-C (<70 mg/dL) was not associated with survival rates over time in the HD group (*p* = 0.503; Figure 2). 

### 3.4. Predictors of Mortality after ACS in HD Group 

As mentioned above, mortality rates were significantly higher among HD vs. control patients both at 30-day and 1-year mortality after ACS, and rates of statin treatment were comparable between HD and control group both before and after ACS hospitalization. There was also no correlation between LDL-C levels and mortality rates (r = 0.1, *p* = 0.321). Among HD patients, 1-year survival after ACS was positively associated with the nutritional parameters of serum albumin (r = 0.368, *p* < 0.001) and total cholesterol (r = 0.185, *p* < 0.018). Survival was inversely correlated with the inflammatory markers CRP (r = −0.348, *p* < 0.001) and NLR; (r = −0.181, *p* = 0.019). Dialysis vintage was not significantly correlated with survival after ACS (r = −0.161, *p* = 0.125). 

Multivariate analysis demonstrated that heart failure was the only significant predictor for 1-year mortality (OR 2.8, *p* = 0.002, Table 4). Low LDL-C (<70 mg/dL) at ACS hospitalization did not predict 1-year mortality in the HD group. 

## 4. Discussion

This retrospective analysis found increased mortality among HD patients presenting with ACS compared to a control group matched for age and sex, yet with a normal eGFR. This was despite numerically lower median LDL-C levels at presentation and similar rates of statin prescription, both at presentation and at discharge. Moreover, a significant proportion of HD patients had LDL-C < 70 mg/dL; however, this parameter did not affect survival among the HD group at both 30-day and 1-year time points.

The finding of increased mortality among HD patients after ACS hospitalization is not surprising; it is well-established as cardiovascular disease is a leading cause of death in this population. Cardiovascular pathophysiology in CKD patients consists of traditional risk-factors (diabetes, dyslipidemia, hypertension), as well as mediators promoting vascular calcification [29]. For example, reduced kidney function and albuminuria are associated with an increase in cardiovascular risk and are independent of these traditional risk factors. In fact, there is an inverse non-linear relationship between eGFR and cardiovascular events, with HD patients experiencing 10- to 30-fold higher CV mortality risk [2,30].

Qualitative abnormalities in serum lipids associated with renal disease rank high among the factors responsible for accelerated atherosclerosis [6]. Thus, the increased prevalence of CVD in patients with ESKD is not necessarily due to elevated LDL-C. More frequently, chylomicrons remnants and high-density lipoprotein deficiency and dysfunction are involved in the increased atherosclerosis observed in ESKD patients. Dyslipidemia and low grade inflammation are also caused by chronic kidney disease [31]. In patients with impaired kidney function, lipid profiles become atherogenic, partly due to defective HDL-C function and excessive oxidation of LDL-C [32]. Mechanisms responsible for systemic inflammation in CKD are incompletely understood. However, increased production of inflammatory mediators has been attributed to raised oxidative stress and accumulation of post-synthetically modified proteins and toxins that are cleared with normal renal function [33].

While older observational studies showed improved survival in HD patients receiving HMG-CoA reductase inhibitors (statins), recent randomized controlled trials yielded conflicting results [24,25,26]. The role of PCSK9 inhibitors in this population has not yet been elucidated. Major prospective studies on patients not undergoing percutaneous coronary intervention (PCI) or upon presentation with ACS have examined the impact of decreased LDL-C on cardiovascular outcomes in HD patients. In the AURORA trial [26], in which 2776 HD patients were divided into a placebo group and a treatment group receiving 10 mg/day rosuvastatin, the combined outcome of cardiovascular death, non-fatal MI, or non-fatal cerebrovascular disorder was not significantly reduced in the treatment group. Similarly, the 4D study [24], which consisted of 1255 HD patients with diabetes mellitus, did not demonstrate a significant reduction in major cardiovascular endpoints in patients receiving 20 mg/day of atorvastatin.

The SHARP study [25] included 9720 patients with CKD (33% on HD). After 5 years of follow-up, combination therapy of simvastatin plus ezetimibe reduced the incidence of major cardiovascular outcomes for the whole cohort. The authors postulated that the failure to achieve significance in the AURORA and 4D studies might derive from the much smaller numbers and proportions of modifiable cardiovascular events in those studies. Of note, the subgroup of HD patients in the SHARP study showed no benefit with treatment. It was mentioned that the study did not have sufficient power to assess the effects on MACE separately in dialysis patients.

In contrast to those studies, information regarding factors related to adverse outcomes, including baseline LDL-C levels, in HD patients presenting with ACS is limited, despite the fact that ACS treatment has progressed significantly in the current era with the use of anti-aggregates, anti-diabetics, and lipid-lowering drugs. The Collaborative Atorvastatin Diabetes Study (CARDS) reported decreased cardiovascular death among patients with type 2 diabetes mellitus treated with atorvastatin in the absence of marked renal insufficiency [34]. However, CKD is still associated with short- and long-term adverse outcomes in patients with cardiovascular disease. While shared risk factors and a sustained inflammatory state may explain this increased risk, guidelines recommending therapies, including statin treatment, are still underutilized [35].

Nagata et al. [36] evaluated factors associated with MACE in HD patients undergoing PCI without ACS compared to a control group of non-HD patients. It was shown that the risk of MACE in HD decreased as LDL-C increased, in contrast to the control group. However, as mentioned, the trial consisted of patients undergoing elective PCI.

Shechter et al. [37] assessed the impact of statin therapy on patients presenting with ACS. They found an inverse correlation between eGFR and statin prescription at discharge. In addition, CKD patients not prescribed statins experienced higher mortality rates. Of note, only 6% of the study population had an eGFR < 30 mL/min, and it was not mentioned whether HD patients were included in the study. In addition, LDL-C levels at admission and the rate of statin usage prior to hospitalization were not assessed.

In our study, more than 50% of the participants were HD patients, potentially enabling better assessment of this specific population. Our analysis showed similar rates of statin administration, both at admission and discharge, between HD patients and the control group. Moreover, median LDL-C levels were lower in the HD group. Low levels of serum total cholesterol in subjects on HD may indicate the presence of malnutrition and are often associated with hypoalbuminemia. Hypocholesterolemia is common in chronic dialysis patients, yet its mechanisms are not well delineated. Cytokinemia related to impaired removal of substances or to exposure to occult chronic inflammation may be involved in the pathogenesis [38]. Declining serum cholesterol levels in HD patients may be an indicator of either declining health or metabolic, dietary, and other clinical abnormalities similar to those associated with old age [39]. Iseki et al. [13] demonstrated a significant correlation between serum cholesterol and albumin levels. In fact, high serum cholesterol levels were an independent predictor of death in a subgroup of patients with relatively high albumin levels. However, a significant proportion of hypocholesterolemic patients in their study had elevated CRP levels. It has been suggested that the paradoxical association between lipid levels and outcomes in HD patients results from either reverse causation, in which advanced cardiovascular disease leads to inflammation, wasting, and lower cholesterol levels, or a confounding effect resulting in lower cholesterol levels and increased mortality [40]. Whereas LDL-C in our cohort did not seem to play a crucial role in the outcomes of HD patients hospitalized due to ACS, our results emphasize the importance of inflammation and malnutrition in their prognosis. Previous trials [13,22] have demonstrated that in the absence of inflammation, the relationship between total cholesterol level (not LDL-C) and mortality in HD patients is the same as in the general population. In a post hoc analysis of the 4D study, Krane et al. [40] examined the combined significance of LDL-C and CRP (as a marker of inflammation) in predicting cardiovascular outcomes among HD patients. They concluded that CRP level is more important than LDL-C in predicting the risk of death and cardiovascular events in those patients. In addition, low albumin, a marker of inflammation and malnutrition, has been shown to increase susceptibility to infection and death in HD patients [41].

Recently, NLR has emerged as a surrogate marker for systemic inflammation in CKD and HD patients [23,42,43]. It is a cost-effective and simple parameter, allowing for the easily assessment of the inflammatory status of a subject, especially in settings where CRP is not measured routinely [44]. It has been shown to be a strong and independent predictor of cardiovascular severity and mortality in maintenance HD patients [45]. Erdem et al. [46] also reported the usefulness of NLR in predicting short-term mortality among patients receiving HD in hospital settings. In a study assessing the relationship of NLR with nutritional markers and outcomes in HD patients, Diaz et al. [23] demonstrated that low NLR levels were associated with decreased hospitalization rate. Although their trial did not reach statistical significance for reduced mortality rates related to low NLR levels, no fatal outcomes occurred in the lowest NLR quartile, while all deaths registered during the study period occurred in the higher NLR quartile. These results support those achieved in other studies. Although baseline NLR did not differ between the control group and the HD group in our study, survival following ACS was significantly associated with increased NLR, thus supporting the findings of previous studies.

We believe our study contributes to the literature by specifically examining the impact of LDL-C at admission with ACS in a large group of HD patients compared to a control group hospitalized at the same time and at the same institute. We also showed that the use of statins was not different between the groups at both hospital admission and discharge. This parameter has important clinical implications since according to current guidelines [47,48], statins are commonly prescribed at discharge following ACS. Our results, in concordance with previous studies, emphasize the importance of nutritional and inflammatory parameters, including albumin, total cholesterol, CRP, and NLR, in predicting clinical outcomes of HD patients following diagnosis of ACS. Thus, nephrologists and healthcare providers of dialysis patients should make every effort to improve the nutritional status of these patients. Since our study did not demonstrate a survival difference related to LDL-C levels above or below 70 mg/dL in HD patients, it would have been interesting to evaluate whether inflammatory and nutritional markers, including CRP, NLR, total cholesterol, and serum albumin, differed between these two groups, thus affecting clinical outcomes. Unfortunately, due to its retrospective design, such a causative association could not be determined in our study.

Our study also had several limitations inherent to its retrospective design and use of electronic databases. Information regarding statin treatment was based on admission and discharge letters. We did not perform regular follow-up on the continuation of statin treatment following discharge in the study population, and we did not differentiate between different statin types or dosages. Since attending nephrologists frequently withhold statin treatment in HD patients, this could potentially influence final outcomes. We also did not collect data regarding glycemic control and treatment or the prescription of other cardioprotective agents following ACS, such as beta blockers, renin-angiotensin system inhibitors (RASI), and sodium glucose 2 transporter (SGLT-2) inhibitors, which are associated with improved survival following ACS. Since a substantial portion of these drugs are either under-used or contraindicated in hemodialysis patients, this could affect prognosis in the study group.

Nevertheless, we believe our results are important since they highlight the value of regular surveillance of inflammatory and nutritional parameters in HD patients presenting with ACS.

## 5. Conclusions

Our study found decreased one-year survival in HD patients presenting with ACS compared to a group of non-dialysis patients with preserved kidney function despite a lower median LDL-C level at diagnosis and similar usage of statin treatment both at diagnosis and discharge. Among the HD group, survival was mainly related to easily measured inflammatory and nutritional parameters. Frequent clinical surveillance is warranted in HD patients at increased risk for mortality following ACS.

## Figures and Tables

**Figure 1 medicina-59-01312-f001:**
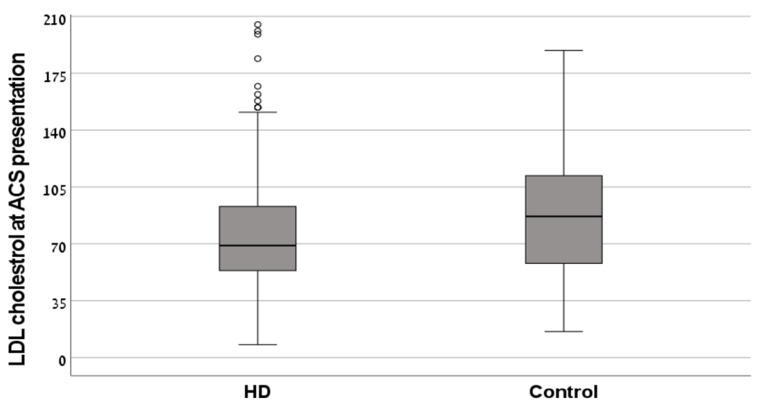
Box plot of LDL-C levels at acute coronary syndrome (ACS) presentation. Box-plot describing median, 25th, and 75th percentile, and minimum and maximum levels of LDL-C (mg/dL) at ACS presentation. The median LDL-C level at ACS presentation was 69 in the HD group and 87 in the control group.

**Figure 2 medicina-59-01312-f002:**
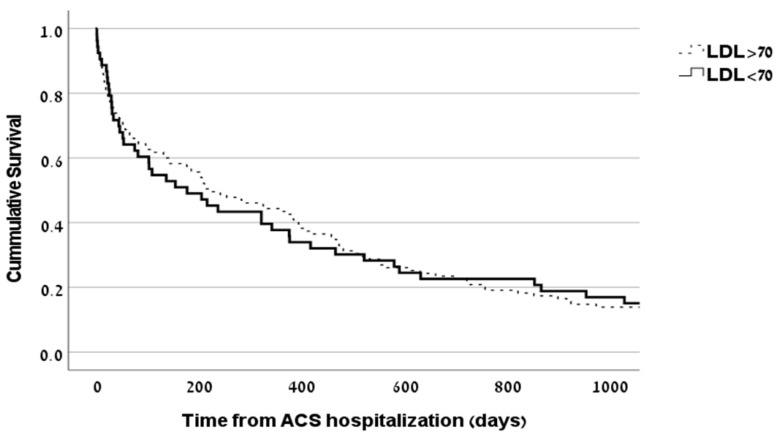
Survival rate in HD group according to LDL-C level. Kaplan–Meier survival curve did not demonstrate differences in survival of HD patients following ACS when comparing by LDL-C (*p* = 0.503). HD, hemodialysis; ACS, acute coronary syndrome; LDL-C, low density lipoprotein cholesterol.

**Table 1 medicina-59-01312-t001:** Baseline demographic and clinical characteristics of study groups.

Variable	HD Group	Control Group (N = 103)	*p*-Value
	(N = 246)		
Age, years; mean ± SD	72.4 ± 10.5	71.3 ± 11.8	0.68
Male sex, n (%)	168 (68.3)	66 (64.1)	0.45
Diabetes, n (%)	159 (64.9)	40 (40.8)	<0.01
Ischemic heart disease, n (%)	175 (71.1)	44 (44.9)	<0.01
Heart failure, n (%)	83 (33.7)	12 (12.2)	<0.01
Hypertension, n (%)	231 (93.9)	67 (68.4)	<0.01
Previous stroke, n (%)	53 (21.5)	9 (9.2)	<0.01
Atrial fibrillation, n (%)	43 (17.6)	8 (8.2)	0.03
Peripheral vascular disease, n (%)	57 (23.2)	6 (6.2)	<0.001
**Laboratory tests at admission for ACS hospitalization**			
WBC (K/microl)	9.8 ± 7.9	10.4 ± 5.6	0.48
Hemoglobin (g/dL)	10.4 ± 1.7	12.7 ± 2.1	<0.01
Platelets (K/microl)	234 ± 110	246 ± 87	0.34
Neutrophil to lymphocyte ratio	9 ± 10.6	7.4 ± 10.9	0.22
C-reactive protein (mg/dL)	7.9 ± 17.5	2.6 ± 4.7	0.01
Serum creatinine (mg/dL)	5.2 ± 11.8	1.1 ± 1	<0.01
Urea (mg/dL)	118.9 ± 60.5	43.7 ± 18.1	<0.01
Albumin(g/dL)	3.4 ± 2.3	3.7 ± 0.5	0.71
Total cholesterol (mg/dL)	153.1 ± 58.1	168 ± 55.2	0.03
HDL-C (mg/dL)	37.9 ± 14.3	40.6 ± 11.1	0.21
Triglycerides (mg/dL)	147.5 ± 80.2	157.2 ± 97.9	0.46
LDL-C (mg/dL)	77.2 ± 41.8	87.4 ± 36.1	0.11

HD, hemodialysis; HDL-C, high density lipoprotein cholesterol; LDL-C, low density lipoprotein cholesterol.

**Table 2 medicina-59-01312-t002:** LDL-C and statin treatment in study groups.

Variable	HD Group	Control Group	*p*-Value
	N = 246	N = 103	
LDL-C (mg/dL), mean ± SD	77.2 ± 41.8	87.4 ± 36.1	0.11
LDL-C < 70 mg/dL, N (%)	80 (32.5)	16 (15.5)	<0.01
Statin treatment before, N (%)	175 (71.1)	61 (59.2)	0.14
Statin treatment after, N (%)	212 (86.2)	84 (81.6)	0.92

LDL-C, low density lipoprotein cholesterol.

**Table 3 medicina-59-01312-t003:** Odds ratio for mortality in study groups.

Coefficients	Odds Ratio	95% Confidence Interval		*p*-Value
		Lower	Upper	
**30-day mortality**				
Control group	Reference			
HD group	5.2	1.8	15	<0.01
**1-year mortality**				
Control group	Reference			
HD group	3.4	1.9	6.1	<0.01

HD, hemodialysis. Low LDL-C (<70 mg/dL) did not predict the 30-day mortality rate in the HD group (*p* = 0.995); the 30-day mortality was 17.5% for both HD patients with LDL-C < 70 mg/dL (14/80) and for those with LDL-C ≥ 70 mg/dL (29/166).

**Table 4 medicina-59-01312-t004:** Multivariate analysis model for 1-year mortality in HD group (n = 246).

Coefficients	Odds Ratio	95% Confidence Interval		*p*-Value
		Lower	Upper	
Female sex	0.907	0.489	1.681	0.756
Diabetes	0.783	0.436	1.406	0.413
Ischemic heart disease	1.633	0.865	3.083	0.13
Heart failure	2.765	1.468	5.208	0.002
Previous stroke	0.628	0.316	1.247	0.183
Atrial fibrillation	1.437	0.678	3.044	0.344
Peripheral vascular disease	0.748	0.38	1.469	0.399
LDL-C < 70 (mg/dL)	1.175	0.658	2.098	0.587

## Data Availability

All data analyzed during this study are included in this article. Further enquiries can be directed to the corresponding author.

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
