# Peer review of "The Clinical Significance of LDL-Cholesterol on the Outcomes of Hemodialysis Patients with Acute Coronary Syndrome"

_medicina, 2023, doi:10.3390/medicina59071312_

Round 1

Reviewer 1 Report

line 86: you should add "(ADMA)" after "asymmetric dimethylarginine"

Table 3= Are you sure about Rsquare=1? Can you report the residual analysis? furthermore, linear regression does not analyse curves, it gives an equation of a straight line. 

Which variable do you analyse through One-way ANOVA e which do you analyse through the Kruskas-Wallis test? yiu report "*One way ANOVA with Bonferroni correction, **Chi squared test, ***Kruskal Wallis test" but I do not see them in the table.

Author Response

Author response:  Thank you for the opportunity to resubmit our manuscript. We revised " Statistical analysis" section in our manuscript as the reviewer suggested. (lines 131-142)

Reviewer 2 Report

Dear authors,

Chronic kidney disease remains a public health problem today, being associated with a marked increase in the incidence of cardiovascular disease mortality. The interest of nephrologists and others is to find biomarkers that must accurately correlate with cardiovascular risk in the CKD patient, especially on dialysis. In this sense, the present paper could support the need to monitor LDL cholesterol as an additional cardiovascular risk factor in patients with CKD/ESRD, along with inflammation and malnutrition.

Regarding the article:

- The abstract needs to be written as a single paragraph and should adhere to the format of structured abstracts, incorporating the following headings: 1) Background and Objectives: Situate the addressed question within a broader context and emphasize the study's purpose; 2) Materials and Methods: Provide a brief overview of the primary methods or treatments employed. Include pertinent preregistration numbers and specify the species and strains of any animals involved; 3) Results: Concisely summarize the main findings of the article; and 4) Conclusion: Clearly state the main conclusions or interpretations derived from the study.

- the introduction could be extended, as could the discussion section.

- the number of references is, in my opinion, insufficient for the discussed subject.

Good luck!

Author Response

Author response:  Thank you for the opportunity to resubmit our manuscript and for your valuable comments. Please find our response below:

  • We rearranged the abstract section according to the format required (lines 15-39).
  • We extended the introduction, as well as the discussion sections, as requested. (introduction section : lines 48-73 ; Discussion section lines 250-257 ; 265-273 ; 327-338 ; 355-372; 399-406).
  • The reference section has been extended from 33 references to 48 references in the revised manuscript to support the extended Introduction and Discussion sections, as suggested.

Reviewer 3 Report

The aim of this article is to investigate the clinical significance of LDL-Cholesterol on the outcomes of hemodialysis patients with acute coronary syndrome.  There are previous studies looking for the association between cholesterol level and mortality in dialysis patients.  However, in this manuscript is reported the effect of LDL cholesterol in hemodialysis patient with ACS.  Although the experimental design is adequate, there are few concerns in the presentation and the interpretation of the results. 

In the abstract, line 32.  It should be clarified that better nutrition and less inflammation were associated with survival of HD patients after ACS.

Page 7, line 202.  It is described survival was inversely correlated with inflammatory markers.  However, only c-reactive protein was determined.    

Although it was not the aim of the study, did the authors analyze if other cardioprotective drugs were used in these patients?  It is known that these drugs are underutilized in this at-risk population and can be a factor to affect the prognosis in these HD patients.

Author Response

  1. In the abstract, line 32.  It should be clarified that better nutrition and less inflammation were associated with survival of HD patients after ACS.

Author response: 

Thank you for noticing this important point. We emphasized that point in the "conclusion" section of the abstract (lines 38-39).

  1. Page 7, line 202.  It is described survival was inversely correlated with inflammatory markers.  However, only c-reactive protein was determined. 

Author response: 

Thank you for bringing this important point to our attention. Neutrophil-to-Lymphocyte ratio (NLR) is a simple biomarker of inflammation that was previously shown to predict cardiovascular and all-cause mortality among hemodialysis patients, and was associated with nutrition and inflammation parameters in HD patients, in several studies. In our study, we did use the NLR as a marker of inflammation but unfortunately did not discuss it appropriately throughout the paper. We therefore added the value of the novel biomarker of NLR, as well as inflammatory process to the introduction section (lines 63-66 ; 71-73) ; to the "methods" section (lines 127-128) ; we added the correlation of NLR to outcomes to the Results section (lines 228-229) and discussed it's important role as an inflammation marker in hemodialysis patients in the Discussion section as well (lines 355-372).

  1. Although it was not the aim of the study, did the authors analyze if other cardioprotective drugs were used in these patients?  It is known that these drugs are underutilized in this at-risk population and can be a factor to affect the prognosis in these HD patients.

Author response: 

Unfortunately, we did not recorded data regarding these drugs. However, we absolutely agree with this important comment and added this limitation to the Discussion paragraph (lines 399-406).

Reviewer 4 Report

This interesting paper examines the role of LDL-cholesterol on mortality in hemodialysis patients with acute coronary syndrome that is rarely assessed before.

Comments:

Line 31 Despite -> despite

When abbreviation is used, it should follow the full version of a term first appears in the article. For instance, "hemodialysis" appear several times in the article however abbreviation appear only in line 77.

Line 86 the abbreviation (ACS) has been provided in line 70.

Line 92-93 acute coronary syndrome -> ACS

Line 101 who had "anth" eGFR >60 ml/min ?

Line 197 Statin -> statin

Line 200 mortality rates (r=0.1, p=0.321)  -> full stop.

Line 204 what is "NLR"

Table 4 why "hypertension" is not present in this table ? Nutritional and inflammatory parameters were significantly correlated with survival, why are they not included in multivariate analysis for 1-year mortality ?

Also table 4, it shows that diabetes was not predictor of mortality. Previous studies have demonstrated glycemic control is associated with mortality both in general population and diabetic hemodialysis patients. [Circulation. 2009;120:2421–2428] [Diabetes. 2012 Mar; 61(3): 708–715] What are HbA1c levels of diabetic patients in both groups ?

Author Response

-Line 31 Despite -> despite.

-When abbreviation is used, it should follow the full version of a term first appears in the article. For instance, "hemodialysis" appear several times in the article however abbreviation appear only in line 77.

- Line 86 the abbreviation (ACS) has been provided in line 70. Line 92-93 acute coronary syndrome -> ACS

- Line 101 who had "anth" eGFR >60 ml/min ?

- Line 197 Statin -> statin

- Line 200 mortality rates (r=0.1, p=0.321)  -> full stop.

Author response:  Thank you for the opportunity to resubmit our manuscript. We have corrected the above comments.

Line 204 what is "NLR"

Author response: 

We absolutely agree with this important remark. Neutrophil-to-Lymphocyte ratio (NLR) is a simple biomarker of inflammation that was previously shown to predict cardiovascular and all-cause mortality among hemodialysis patients, and was associated with nutrition and inflammation parameters in HD patients, in several studies. In our study, we did use the NLR as a marker of inflammation but unfortunately did not discuss it appropriately throughout the paper. We therefore added the value of the novel biomarker of NLR, as well as inflammatory process to the introduction section (lines 63-66 ; 71-73) ; to the "methods" section (lines 127-128) ; we added the correlation of NLR to outcomes to the Results section (lines 228-229) and discussed it's important role as an inflammation marker in hemodialysis patients in the Discussion section as well (lines 355-372).

Table 4 why "hypertension" is not present in this table ? Nutritional and inflammatory parameters were significantly correlated with survival, why are they not included in multivariate analysis for 1-year mortality ?

Author response: 

Table 4 show results of multivariate regression analysis model for predicting 1-year mortality among HD patients. In our cohort, as well as in other cohort, hypertension diagnosis was prevalent among HD patients. However, since hypertension may reflect volume overload and not necessarily essential hypertension among HD patients, and mainly due to high variability in BP measurements (before, during and after hemodialysis treatments), we believe adding hypertension to the model can lead to a bias and misinterpretations. However,  Heart failure remained significant and Hypertension was not significant predictor of mortality when we added it to model:

p value

95% C.I.

OR

Upper

Lower

0.252

1.263

0.411

0.720

DM

0.795

3.631

0.372

1.163

Hypertension

0.000

4.795

1.550

2.726

heart failure

0.079

3.207

0.939

1.735

IHD

0.155

1.206

0.307

0.609

s/p stroke

 In our study, we used CRP and NLR as inflammatory markers (continuous variables) and assessed their impact using one way ANOVA and correlation tests. In the above model, we included only nominal parameters and therefore did included age, CRP and NLR in this model. 

Also table 4, it shows that diabetes was not predictor of mortality. Previous studies have demonstrated glycemic control is associated with mortality both in general population and diabetic hemodialysis patients. [Circulation. 2009;120:2421–2428] [Diabetes. 2012 Mar; 61(3): 708–715] What are HbA1c levels of diabetic patients in both groups ?

Author response: 

Laboratory tests that were included in this paper were taken at the ACS presentation. Unfortunately, HBa1C levels were not assessed during the hospitalization and cannot be added. We added this important note to the limitation paragraph (lines 399-406). 

Round 2

Reviewer 1 Report

Requested reviews have been applied. 

Reviewer 2 Report

Thank you for rearranged and verified the article according with reviewers recommendations. I don't have any other questions.